# INTRA: Interleaved Non-contiguous Token spaRse Attention

## Abstract

Transformers achieve strong performance across modalities but are bottlenecked by the quadratic cost of attention. Sparse attention reduces this cost, yet existing methods either incur high runtime overhead (dynamic patterns) or lose GPU efficiency due to blockwise masking (static patterns). We introduce **INTRA**—**I**nterleaved **N**on-contiguous **T**oken spa**R**se Attention—a token-level sparse attention kernel that decouples memory loading from computation. INTRA preserves blockwise access for GPU efficiency while flexibly supporting non-contiguous token sparsity, eliminating mix-block overhead. To ensure global information exchange, INTRA interleaves complementary static patterns across layers. We further propose the **ISPD** Principle, a general guideline for constructing hardware-efficient sparse patterns. On `FLUX.1-dev`, INTRA reduces 2K image generation latency from $66s \rightarrow 43s$ with no quality loss after LoRA self-distillation. On LLaMA-3.1-8B, INTRA matches existing sparse attention methods on the SCROLL benchmark with sequence length up to 16K, demonstrating its scalability.

## 1 Introduction

Transformers excel across domains such as NLP and image generation, but their attention cost scales quadratically with sequence length, creating a bottleneck for long contexts and high-resolution generation. For instance, `FLUX.1-dev` Team (2024) requires nearly a minute to produce a 2048×2048 image on a single A100, making generating high-quality images in real time impractical.

Sparse attention has been widely explored to reduce the computation overhead of full attention Liu et al. (2025); Jiang et al. (2024). Sparse attention kernels can be broadly categorized into **dynamic** and **static** designs. Dynamic sparse kernels flexibly adapt the sparse pattern at runtime and can be directly applied to pretrained models without finetuning. However, their runtime flexibility comes at the cost of higher memory and computation overhead, leading to slower inference. In contrast, static sparse kernels fix the sparsity pattern in advance. While this requires model finetuning to adapt to the kernel, it results in significantly higher efficiency at inference time. We focus on static kernels in this work, aiming to design them in a GPU-friendly manner while still enabling rich token interactions.

One limitation of existing sparse attention methods is that they mainly preserve local patterns, preventing tokens from accessing the global context. Since attention can be viewed as message passing, restricting information flow to local neighborhoods weakens the model's ability to capture long-range dependencies. We argue that effective sparse attention must maintain complete information propagation across all tokens.

Prior works address this by carefully crafting patterns. For example, CLEAR Liu et al. (2025) adopts circular local sparsity, and NATTEN Hassani et al. (2023) uses sliding windows. However, these fine-grained designs break the blockwise structure required for GPU efficiency. As shown in Figure 2, even when only part of a block is needed, the entire block must still be computed and then masked, adding overhead and diminishing real speedups compared to theoretical FLOP reductions.

We introduce **INTRA**—**I**nterleaved **N**on-contiguous **T**oken spa**R**se Attention. INTRA applies various sparse patterns, including potentially non-contiguous ones, across different attention layers of a model. Although such non-contiguous patterns are often overlooked in existing sparse attention design due to concerns about non-locality and memory cost, we find that these are not the primary obstacle. The critical requirement is that cohorts of query tokens consistently attend to the same key subset

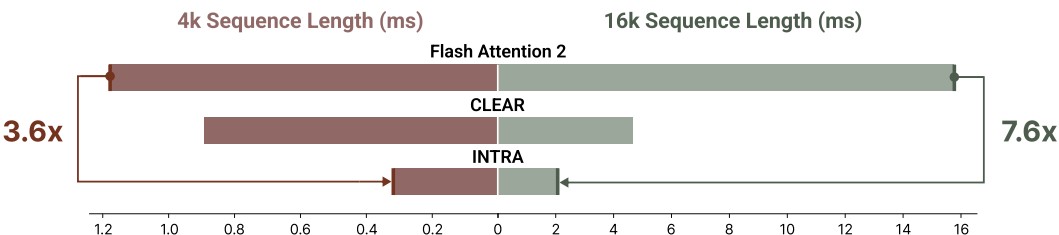

Figure 1: The figure reports the relative speedup achieved by INTRA compared to CLEAR and FlashAttention 2 on 4K and 16K input lengths, evaluated on an A100 GPU. Note that the axis density differs between the left and right sides of the figure to accommodate varying inference times.

within a sparse pattern (i.e., only "intra" connections are allowed). INTRA enforces this property, preserving blockwise memory access (and thus compatibility with FlashAttention) while avoiding mix-block overhead and enabling more expressive sparse structures. The tradeoff of this efficiency is that information flows only locally in a layer. To recover global connectivity, INTRA interleaves complementary static patterns across layers, ensuring that every token can directly or indirectly attend to all others. (Illustration: think of two interwoven lattices that together span the entire sequence.) To generalize this design, we propose the Intra Sparse Pattern Design (ISPD) Principle, a guideline for constructing general hardware-efficient sparse patterns that extends beyond token-level designs.

Our contributions are: 1. A token-level sparse attention kernel that decouples the memory loading from the attention calculation. 2. Interleave different patterns to ensure full information exchange of all tokens while maintaining sparsity. 3. The ISPD Principle, a unified framework for designing efficient sparse patterns, covering both token-wise and sliding window sparsity.

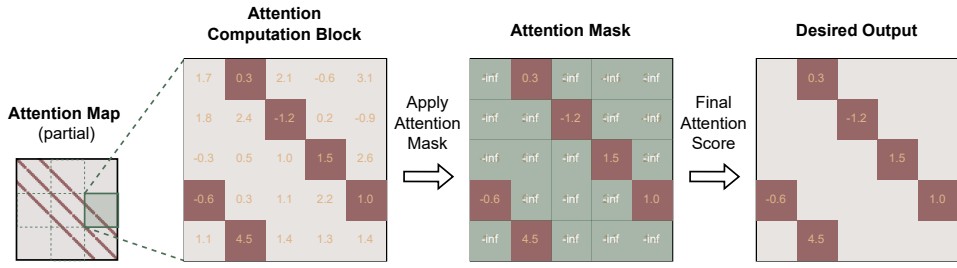

Figure 2: Low-level implementation of blockwise sparsity for handling mixed blocks. A mix block is the block that partially preserves attention scores, as shown on the left side of the figure.

We apply INTRA to both image and text generation models, achieving significant speedups without quality degradation. On the `FLUX.1-dev` image model, INTRA matches dense attention quality while reducing 2K image generation from 66s $\rightarrow$ 43s on a single A100 after LoRA self-distillation. On LLaMA 3.1 8B, INTRA achieves accuracy comparable to dense, MInference Jiang et al. (2024), and XAttn Xu et al. (2025) on the long context SCROLL Shaham et al. (2022) benchmark that scales to 16K tokens with better efficiency. We hypothesize that INTRA could deliver even greater speedups on longer sequences. However, because it employs a static-pattern sparse attention mechanism, fine-tuning at such extended sequence lengths exceeds our computational budget. Notably, a 16K context represents a fairly standard long-context setting for both image and text tasks, and aligns with the sequence length used in video generation (e.g., Wan 1.3B Wan et al. (2025) at 480p).

## 2 RELATED WORKS

### 2.1 STATIC SPARSE ATTENTION

Sparse attention mechanisms reduce the quadratic complexity of full attention by limiting token-to-token interactions, making them crucial for scaling transformers to long sequences. Static sparse

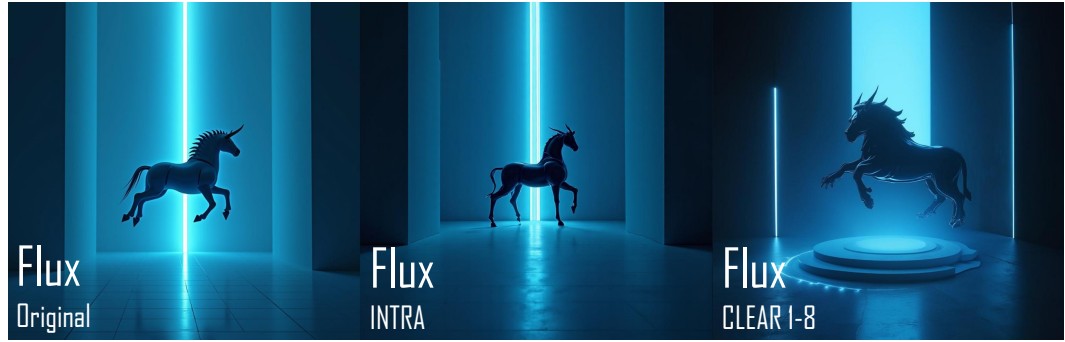

Figure 3: Visual comparison on `FLUX.1-dev` with different implementations: (left to right) Original FlashAttention 2, INTRA (our method), and CLEAR. As shown, INTRA maintains image quality. Please refer to Appendix I for more comparison examples.

attention adopts fixed, input-independent patterns, enabling predictable computation. For instance, sliding window attention Jiang et al. (2023) restricts each token to attend only to a fixed local neighborhood, effectively capturing local dependencies. StreamingLLM Peng et al. (2024) proposes a memory mechanism that retains only the initial and most recent tokens, ensuring bounded latency and memory usage in long-context settings. Dilated attention Hassani & Shi (2022) introduces periodic strides to expand receptive fields while maintaining sparsity. Different works adopt sliding window attention to image-related tasks. NATTEN Hassani et al. (2023) restricts each query token to attend a 2D neighborhood region. CLEAR Liu et al. (2025) reduces the attention complexity of diffusion transformers by employing a fixed local window. To improve context coverage, mixed sparse patterns combine different attention types. Notably, BigBird Zaheer et al. (2020) combines global tokens, local windows, and random connections to enable sparse yet expressive attention. Similarly, Longformer Beltagy et al. (2020) integrates global and windowed attention mechanisms.

## 2.2 DYNAMIC SPARSE ATTENTION

In contrast, dynamic sparse attention adapts its patterns based on input content or model state, aiming to allocate compute selectively to salient regions. MInference Jiang et al. (2024) and FlexPrefill Lai et al. (2025) speed up prefill by selecting sparse patterns per attention head, although the cost of selection remains a bottleneck. xAttention Xu et al. (2025) employs anti-diagonal scores to identify and prioritize important token blocks dynamically. Native Sparse Attention (NSA) Yuan et al. (2025) learns sparse attention masks directly through supervision or reinforcement learning, enabling flexible adaptation to diverse long-context inputs. MoBA Lu et al. (2025a) leverages a Mixture-of-Experts routing scheme to assign attention patterns based on input characteristics, balancing accuracy and efficiency. H2O Zhang et al. (2023) introduces a hierarchical routing strategy to dynamically switch between coarse and fine-grained attention.

## 2.3 OTHER IMAGE ACCELERATION METHODS

Beyond sparse attention, various token reduction strategies have been proposed to accelerate vision transformer inference by reducing the effective sequence length. Token pruning methods eliminate less informative tokens during inference. Dynamic approaches such as Dynamic Token Sparsification Rao et al. (2021) or SPViT Kong et al. (2022) assess token importance via learned scoring networks or attention distributions. Token merging offers an alternative that avoids hard removal. ToMe (Token Merging) Bolya et al. (2023) merges similar tokens based on feature proximity. ToMA Lu et al. (2025b) extends ToMe by applying adaptive merging in both spatial and channel dimensions.

## 2.4 SPARSE VIDEO GENERATION

Extending sparse attention to video generation introduces additional challenges due to the high dimensionality of spatiotemporal data. Video models must capture both local spatial details and long-range temporal dynamics, while keeping computation feasible. STA Zhang et al. (2025) enforces each

Q block to attend to a strict region to improve hardware efficiency, ensuring that the sparse pattern remains well-suited for GPU block-wise calculation. Similarly, Video Swin Transformer Liu et al. (2021) applies shifted window attention across spatial and temporal dimensions, reducing complexity while maintaining temporal coherence. SparseVideo Xi et al. (2025) categorizes heads into spatial and temporal ones, enabling efficient long-sequence modeling in diffusion-based video generation.

## 3  Scatter and Gather Sparse Pattern Design

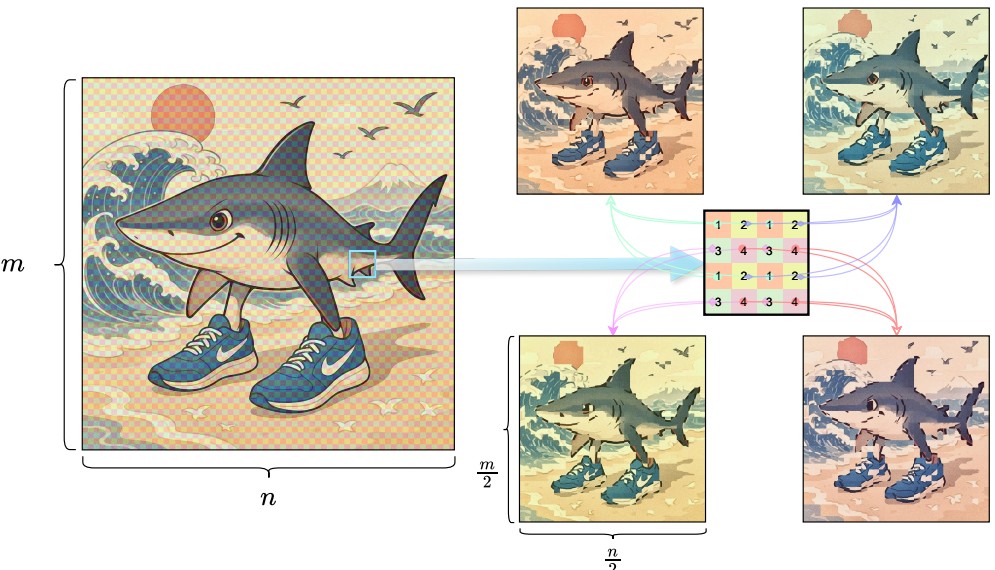

Figure 4: Our intuition comes from the image subsampling. We split image pixels into four groups, and the image formed by a single group can still preserve the global shape and content.

The idea behind our token-wise sparsity stems from the locality of images, and with modifications, the same concept can be applied to language generation. As illustrated in Figure 4, if we divide an image into non-overlapping 2×2 patches and select only one pixel from each patch, we can form a new image that is 4 times smaller in resolution. Despite the lower resolution, the overall structure and content of the image are still recognizable. This suggests that only attending to a subset of tokens may be sufficient to preserve global image information. Based on this intuition, we apply a similar concept to the attention mechanism. We divide the input feature map into non-overlapping patches. Each patch contains a fixed number of tokens. Tokens at the same position across all patches are grouped together, forming multiple token groups. Since we only compute attention within each group, this pattern is referred to as **Scatter** pattern in Figure 5. Suppose the number of groups in the Scatter pattern is $M$. Since attention is now applied within each group, the total number of computations is reduced to $M \times \frac{1}{M^2} = \frac{1}{M}$. This leads to a significant reduction in computation. We also note that **Scatter** pattern is different from a dilated convolution, as the pattern applies to all tokens in the same group and is not restricted to a convolutional local region.

However, this Scatter attention design alone causes a major limitation: tokens in one group cannot access information from other groups, leading to information isolation. Our experiments confirm this issue in Table 5—image generation tasks work poorly when using only the Scatter pattern. To solve this, we introduce the **Gather** pattern. The Gather pattern gathers tokens from all groups to perform attention. Since each token carries the information of all tokens within its group, the Gather pattern enables information sharing between groups.

For the `FLUX.1-dev` model, where each query attends to a fixed-size neighborhood in the Gather pattern (**Gather 0** in Figure 5). In practice, we use a 16×16 region, which spans multiple groups and enables tokens to attend to information outside of their groups. For language models, we apply a different Gather pattern (**Gather 1** in Figure 5). Suppose the Scatter pattern has $G$ groups. The Gather

Figure 5: This figure illustrates the three attention patterns used in INTRA. Scatter and Gather 0 are designed for vision tasks, where query and key tokens are arranged in a 2D feature map format. Gather 1 is used for language tasks and is visualized in a 1D token sequence layout. Due to space constraints, the full token sequence in Gather 1 is split into multiple 1D segments, with curved arrows indicating their consecutive order. We also apply the Scatter pattern to language tasks. For more detailed visualizations of the LLMs' Scatter and Gather 1 patterns, please refer to Appendix A.

1 also has $G$ groups. Each group chooses contiguous $G$ tokens along the input order one by one. This token grouping repeats until all tokens belong to a group. Then, we perform attention within each group. This effectively allows interaction between different groups because each one contains tokens from all groups. By interleaving Scatter and Gather patterns, we ensure that all tokens can exchange information, either directly or indirectly, while maintaining efficient attention computation. In contrast, solely using either pattern will not work in the image task as shown in the Table 5.

## 4 INTRA SPARSE PATTERN DESIGN

In this section, we introduce the Intra Sparse Pattern Design (ISPD) Principle for guiding the hardware-efficient sparse pattern design. The principle is intuitively simple: sparse patterns that have only intra-CQS attention are efficient. We formalize the ISPD Principle using the concept of *Computational Query Sets (CQS)*.

A **Computational Query Set** is a group of query tokens that attend to the same set of key tokens under a sparse attention pattern. Each token in a CQS is associated with four indices:

- $i_g^q$: Global index of the query token in the input sequence
- $i_l^q$: Local index of the query token within its CQS
- $i_g^k$: Global index of the key token in the input sequence
- $i_l^k$: Local index of the key token within the attended key set

We define a sparse attention pattern as hardware-efficient if it satisfies the following three conditions:

1. **Coverage:** The union of all CQSs fully covers the set of query tokens. Each query token can belong to more than one CQSs.
2. **Index Mapping:** There exist two functions $f_q$ and $f_k$ such that, for a given CQS index $i^{\mathrm{cqs}}$,

$$i_g^q = f_q(i_l^q, i^{\mathrm{cqs}}), \quad i_g^k = f_k(i_l^k, i_l^q, i^{\mathrm{cqs}}) \tag{1}$$

These functions map local indices within the CQS to their global positions in the input sequence.

3. **Block Alignment:** Each CQS should contain a number of query tokens approximately divisible by a predefined query block size $B$. In our case, we use $B = 64$ to match the block size used in FlashAttention.

Our proposed Scatter and Gather patterns adhere to the ISPD principle. Each group in the Scatter pattern serves as a CQS. For instance, in the LLM Scatter configuration with 8 groups and an input sequence length of 4096, each CQS comprises 512 tokens. This size is divisible by the block size $B = 64$. The relevant mapping functions $f_q$ and $f_k$ are provided in Appendix C.1. In our proposed Gather 0 pattern, each query block is treated as a CQS. This satisfies the coverage and block alignment conditions. We provide the corresponding $f_q$ and $f_k$ functions in Appendix C.2.

Since attention computation is equivariant, its output changes in the same way we change the query order, regardless of the K/V order. Therefore, if a pattern follows the ISDP principle, we can manipulate the Q order in the attention map to group the tokens in the same CQS together. Then, in each CQS, we can reorder the K in the attention map to gather the computation-needed places together to form the structured rectangular regions as shown in Figure 6.

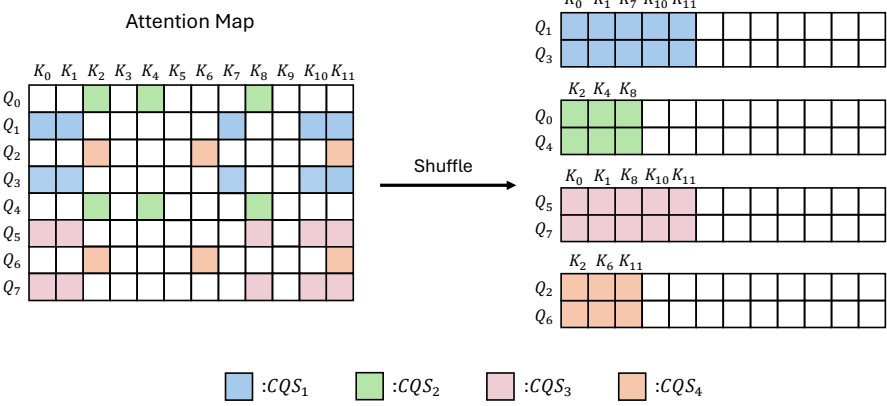

Figure 6: We split the sample attention map into 4 CQSs. For each CQS, we can group the scores in the attention map together to form the rectangular regions.

To explore the full potential of our design, we formulate the problem of general sparse pattern implementation as a constrained optimization problem. The objective is to find the CQSs that minimize the masking steps. Details of this formulation are provided in Appendix D.

## 5 KERNEL DESIGN: DECOUPLING MEMORY LOADING FROM COMPUTATION

We build our kernel by extending FlashAttention. To enable sparse pattern compatibility, we decompose FlashAttention into two stages:

1. Global memory interaction: FlashAttention loads the Q/K/V/O blocks from global memory into shared memory and writes the output tokens back to global memory (Please refer to Appendix B for GPU memory hierarchy background information).

2. Attention computation: Performing softmax and value aggregation with shared memory.

Our kernel design follows the ISPD principle. It selects only the necessary Q, K, and V tokens required by the sparse attention calculation from global memory and stores them contiguously into shared memory. Our design changes the loading unit from blocks to potentially non-contiguous tokens in global memory. Because we place the tokens in the contiguous location in shared memory, the following attention computation remains unchanged and continues to leverage the high performance of FlashAttention. After computation, the attention outputs are written back to their initial, potentially non-contiguous positions in global memory. In comparison, a blockwise implementation could only simulate this behavior by masking out unwanted values after loading large contiguous blocks—an

approach that is less efficient both in memory and compute. Our design ensures that the sparse patterns that do not align with blockwise computation can still be efficiently implemented.

In fact, our custom token loader is a generalized version of the original FlashAttention contiguous block loading. FlashAttention iteratively loads the Q, K, and V blocks according to the input token order to compute outputs. This works well with dense attention but severely limits the flexibility needed to support irregular or token-wise sparse patterns. Token-wise loading can perform block loading by selecting the contiguous tokens inside the block. This flexible loading mechanism allows us to implement many sparse patterns efficiently. Our kernel can only load the neighborhood required by each Q block, making the neighborhood tilling more adaptable.

## 6 EVALUATION

We evaluate INTRA on both the `FLUX.1-dev` Team (2024) image generation model and the LLaMA 3 Touvron et al. (2024) language model. Importantly, INTRA aims to mimic dense attention by allowing full information interaction. In this way, INTRA changes how the model exchanges information between tokens, and therefore, a fine-tuning stage is required. In practice, it is sufficient to use LoRA Hu et al. (2022) adapters rather than full parameter fine-tuning to adapt the model to INTRA. This makes INTRA applicable to existing models without heavy re-training. We hypothesize that models pretrained with INTRA could potentially have better performance.

### 6.1 EXPERIMENTS

**Baseline** INTRA aims at simulating the full information exchange of dense attention. This differs from the majority of previous sparse attention works that use dynamic or static patterns to preserve the partial attention scores. Therefore, not many works are within our comparison scope. For image generation, we chose CLEAR as the baseline method as it requires fine-tuning of the model. It is the previous state-of-the-art static sparse pattern method. Since INTRA also uses static patterns, comparing with CLEAR can test the fine-tuning efficiency of the INTRA interleaved design.

For text generation, we compare with MInference Jiang et al. (2024) and XAttn Xu et al. (2025) on the long context SCROLL benchmark Shaham et al. (2022). We do not test INTRA on video generation tasks due to limited resources. The SCROLL benchmark tasks have a long sequence length up to 16K tokens, which matches the sequence length of the Wan 1.3B video generation at 480p resolution.

**Experiment Setup** We use the same 10K self-generated image-text dataset used by CLEAR to fine-tune `FLUX.1-dev`. Following the convention from CLEAR, we randomly sample 5k COCO Lin et al. (2015) validation set prompts to generate images for evaluation on FID Seitzer (2020), LPIPS Zhang et al. (2018), and CLIPI Radford et al. (2021). For LLaMA 3, we fine-tune and evaluate using the SCROLL Shaham et al. (2022) benchmark. It provides standard train, validation, and test splits. We conduct experiments on the GovReport Huang et al. (2021), Qasper Dasigi et al. (2021), QMSum Zhong et al. (2021), QUALITY Pang et al. (2022), SummScreen-FD Chen et al. (2022), and ContractNLI Koreeda & Manning (2021) tasks. These datasets were selected because their average 16K input sequence lengths fall within our fine-tuning capability, and they provide sufficient validation sets for reliable evaluation.

**Resource Usage and Practicality** All the fine-tuning and evaluations are conducted on NVIDIA A100 GPUs. The `FLUX.1-dev` model fine-tuning takes 43 GPU hours, while LLaMA 3 fine-tuning takes 320 GPU hours. These resource requirements are minimal compared to the cost of full pretraining for these models, demonstrating that INTRA is a practical and efficient sparse solution.

**Results** The speedup comparisons of `FLUX.1-dev` and causal attention speedup are in the Table 1 and Table 2, respectively. For image generation, we compare with CLEAR in both 1024×1024 and 2048×2048 generation settings. The benchmark results of `FLUX.1-dev` are in the Table 3. INTRA and CLEAR scores are close on 1k image generation, while INTRA exceeds CLEAR in 2k image generation. The end-to-end generation time comparison is in the Table 3. INTRA achieves the lowest inference time in both 1k and 2k image generation. For language tasks, INTRA achieves the best speedup in all sequence length cases in Table 2 for single causal attention forward. The end-to-end inference comparison up to 16K and the SCROLL Shaham et al. (2022) benchmark results are in the Table 4. INTRA achieves the lowest latency in all sequence length settings. INTRA can approach the

Table 1: Single full attention speed comparison on A100 in image geneartion. In this table, we interleave Scatter/Gather patterns to get the average latency and speedup.

| Methods | Seq Len | Config | GFLOPS | Latency(ms) | Speedup | FID ↓ |
|---|---|---|---|---|---|---|
| FlashAttention 2 | | - | 260.9 | 1.181 | - | - |
| CLEAR | 4K | r=8 | 63.5 | 0.893 | 1.32× | 11.70 |
| INTRA | | Interleave Scatter/Gather | 50.75 | 0.324 | 3.64× | 11.74 |
| FlashAttention 2 | | - | 3507.9 | 15.764 | - | - |
| CLEAR | 16K | r=8 | 246.2 | 4.625 | 3.41× | 32.46 |
| INTRA | | Interleave Scatter/Gather | 345.5 | 2.056 | 7.67× | 15.67 |

Table 2: Single causal attention speed comparison on A100 in text generation. In this table, we interleave Scatter/Gather patterns to get the average latency and speedup.

| Methods | Seq Len | GFLOPS | Latency(ms) | Speedup |
|---|---|---|---|---|
| FlashAttention 2 | 4K | 137.4 | 0.82 | - |
| INTRA | | 17.2 | 0.16 | 5.1× |
| FlashAttention 2 | 8K | 549.8 | 2.86 | - |
| MInference | | 329.8 | 14.3 | 0.2× |
| XAttn S=16 | | 153.6 | 1.91 | 1.5× |
| INTRA | | 68.7 | 0.51 | 5.6× |
| FlashAttention 2 | 16K | 2199.1 | 10.57 | - |
| MInference | | 1275.4 | 26.42 | 0.4× |
| XAttn S=16 | | 614.2 | 4.22 | 2.5× |
| INTRA | | 274.9 | 1.75 | 6.0× |
| FlashAttention 2 | 32K | 8796.3 | 40.96 | - |
| MInference | | 4398.1 | 51.2 | 0.8× |
| XAttn S=16 | | 2615.1 | 8.03 | 5.1× |
| INTRA | | 1099.5 | 6.5 | 6.3× |

Table 3: End-to-end inference speed comparison between INTRA and CLEAR on 1k and 2k image generation, along with their image quality scores. CLEAR (r=4–8) samples K and V at 1/4 resolution with a radius of 8. We omit results for CLEAR (r=4–8) as its generation time exceeds that of the original dense `FLUX.1-dev` model.

| Method | 1024×1024 Image Generation | | | | 2048×2048 Image Generation | | | |
|---|---|---|---|---|---|---|---|---|
| | FID (↓) | LPIPS (↓) | CLIPI (↑) | Gen Time(s) | FID (↓) | LPIPS (↓) | CLIPI (↑) | Gen Time(s) |
| Original Model | - | - | - | 13.42 | - | - | - | 66.78 |
| CLEAR (r=8) | **11.70** | 49.96 | 0.8995 | 12.85 | 32.46 | 61.62 | 0.8046 | 47.81 |
| CLEAR (r=4-8) | - | - | - | 13.73 | 19.27 | 54.60 | 0.8467 | 52.07 |
| INTRA | 11.74 | **49.46** | **0.9045** | **11.97** | **15.67** | **51.99** | **0.8559** | **43.72** |

performance of the XAttention and MInference after the LoRA fine-tuning on the task Gov Repor, Qmsum, and Summ Screen Fd. INTRA even exceeds in the Quality task.

## 6.2 ABLATION STUDY

To evaluate the effectiveness of INTRA for image generation, we conduct two ablation studies: (1) using Scatter pattern only, using Gather pattern only, or interleaving both; (2) the impact of different numbers of groups in the Scatter pattern when interleaving two patterns. The results are in the Table 5. We further investigate the interleaving frequency, and the experiment details are in the Appendix G

Table 4: End-to-end latency comparison (left) and SCROLL benchmark performance (right). INTRA achieves the lowest latency while maintaining competitive benchmark results.

| Method | Prefill Latency (s) | | | SCROLL Long-context Benchmark | | | | | |
|---|---|---|---|---|---|---|---|---|---|
| | 8k ↓ | 16k ↓ | 32k ↓ | GovReport ↑ | Qasper ↑ | Qmsum ↑ | Quality ↑ | SummFd ↑ | ContractNLI ↑ |
| XAttention | 0.61 | 1.20 | 2.35 | 31.16 | 39.94 | 24.93 | 27.5 | 20.27 | **77.5** |
| MInference | 1.08 | 1.94 | 3.82 | **31.44** | **39.99** | **25.18** | 24.5 | **20.40** | 77.1 |
| INTRA | **0.55** | **1.09** | **2.21** | 27.77 | 39.64 | 24.45 | **29.0** | 19.71 | 75.5 |

Table 5: Ablation study: Left—whether to use Scatter/Gather or both; Right—number of groups used in Scatter during interleave.

| (a) Use Scatter/Gather or both | | | | (b) Number of groups in Scatter | | | |
|---|---|---|---|---|---|---|---|
| | **FID (↓)** | **LPIPS (↓)** | **CLIPI (↑)** | | **FID (↓)** | **LPIPS (↓)** | **CLIPI (↑)** |
| Scatter | 213.47 | 64.91 | 0.67 | 4 groups | **10.99** | **48.29** | **0.908** |
| Gather 0 | 13.23 | 54.10 | 0.88 | 8 groups | 11.75 | 49.46 | 0.905 |
| Interleave 0/1 | **11.75** | **49.46** | **0.90** | 16 groups | 12.85 | 49.88 | 0.896 |

Table 5 (a) presents the results of using the Scatter pattern only, the Gather pattern only, and interleaving both patterns. We observe that using any single pattern leads to significantly worse performance. In contrast, interleaving both patterns achieves the best results across all evaluation metrics. This demonstrates the effectiveness of our interleaving approach, where Scatter and Gather together exchange information among tokens. Table 5 (b) shows the results of varying the number of groups in the Scatter pattern. The 8-group setting achieves a trade-off between 4 groups and 16 groups, which maintains good speedup and causes minor performance degradation.

## 6.3 Overhead and Limitation

INTRA introduces some overhead from computing the mapping functions $f_q$ and $i_g^k$ (local-to-global index conversion), which rely on costly modulo operations executed at each memory access. Additional overhead arises from uncoalesced global memory access: the scatter operations load tokens from non-contiguous addresses, resulting in inefficient transactions. Compared to Gather 0, Scatter suffers slightly lower speedups due to fewer local reads and more dispersed global accesses. Finally, we were unable to evaluate INTRA on video models due to limited computational resources. While smaller video models as Wan 1.3B, share similar sequence lengths (16K) with our studied image and text workloads, state-of-the-art video models require substantially longer sequences. This setting could amplify INTRA's advantages, but we lack enough resources to conduct large-scale video experiments to verify the speedups. We leave it for future work.

## 7 Conclusion

We introduced INTRA for an alternative sparse attention design that supports non-contiguous token access under a blockwise implementation, avoiding mix-block overhead and preserving compatibility with FlashAttention. By enforcing intra-block connections and interleaving complementary static patterns across layers, INTRA enables full information propagation while maintaining efficient memory access. We further proposed the ISPD Principle, a general guideline for hardware-friendly sparse pattern design. INTRA achieves substantial gains in practice: on `FLUX.1-dev`, it reduces 2K image generation from 66s to 43s on a single A100 without quality loss, and on LLaMA 3.1 it matches the Minference and XAttn on the SCROLL long-context benchmark. At this scale, INTRA delivers up to $7.7\times$ (full) and $6.3\times$ (causal) speedups over dense attention. These results show that INTRA makes long-context and high-resolution generation significantly faster and more resource-friendly, paving the way for broader deployment of efficient sparse attention.

## 8 THE USE OF LARGE LANGUAGE MODELS (LLMS)

Large Language Models (LLMs) were used exclusively as auxiliary tools to enhance the writing of this paper, focusing on grammar, phrasing, and stylistic clarity. They did not influence the conception of research ideas, methodological design, execution of experiments, or interpretation of results. All core scientific contributions—including problem formulation, technical development, and validation—were performed solely by the authors.

## 9 REPRODUCIBILITY STATEMENT

Details of the experimental setup, including the models and benchmark tasks, are included in Section 6.1. The detailed configuration of the INTRA index function is in Section C. The INTRA kernel implementation will be made publicly available on GitHub in the future.

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

## A    LLM SCATTER AND GATHER 1 PATTERNS

Illustration of Scatter and Gather 1 Patterns on LLM viewed in 1D token sequence layout.

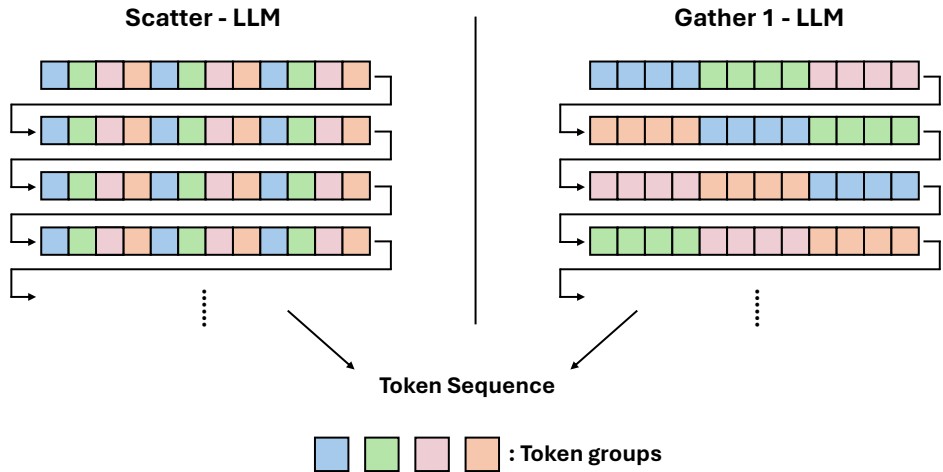

Figure 7: The figure depicts the Scatter and Gather 1 patterns applied to the language model's input tokens, with a grouping factor of 4. Tokens are partitioned into 4 distinct groups, each represented by a different color. Both patterns perform attention within each group.

## B    GPU MEMORY HIERARCHY

NVIDIA GPUs are designed with a hierarchical memory architecture that includes three primary levels: Global memory, Shared memory, and Register. Global memory is a large-capacity, high-latency memory where all data resides initially. Shared memory is a fast, medium-sized memory space shared among threads within the same thread block. Registers are the fastest and smallest memory, used by individual threads for performing computations.

## C    SPARSE PATTERN INDEX FUNCTION

### C.1    SCATTER

We take the LLM Scatter pattern as an example here. The Query block size is 64. The CQS index is $i^{cqs}$. The group number is 8.

$$f_q(i_l^q, i^{cqs}) = i^{cqs} + 8 \cdot i_l^q \tag{2}$$

$$f_k(i_l^k, i_l^q, i^{cqs}) = i^{cqs} + 8 \cdot i_l^k \tag{3}$$

### C.2    GATHER 0

Assume the image feature map has width $W$ and height $H$. The Query block size and Key block size are both $8\times8=64$, and the neighborhood region is $16\times16$. The CQS index is $i^{cqs}$. The group number is 8.

$$f_q(i_l^q, i^{cqs}) = \left\lfloor \frac{i^{cqs}}{(W/8)} \right\rfloor \cdot (8 \cdot W) + (i_l^q \ \% \ 8) \cdot W + (i_l^q \ \% \ 8) \tag{4}$$

$$f_k(i_l^k, i_l^q, i^{cqs}) = \begin{cases} f_q(i_l^q, i^{cqs}) - 4 \cdot W + \frac{i_l^k}{8} \cdot W + i_l^k \ \% \ 8 & \text{if } i^{cqs} \ \% \ (W/8) == 0 \\ f_q(i_l^q, i^{cqs}) - 4 \cdot W - 8 + \frac{i_l^k}{8} \cdot W + i_l^k \ \% \ 8 & \text{if } i^{cqs} \ \% \ (W/8) == (W/8) - 1 \\ f_q(i_l^q, i^{cqs}) - 4 \cdot W - 4 + \frac{i_l^k}{8} \cdot W + i_l^k \ \% \ 8 & \text{else} \end{cases} \tag{5}$$

---

**Algorithm 1:** INTRA Kernel Implementation

---

**Input:** $Q, K, V, O$
**Output:** $O$

Load $Q$, $K$, $V$, $O$ from global memory to shared memory according to $f_q$, $f_k$, $f_v$, $f_o$. $f_q$ and $f_o$ have same formulation, $f_k$ and $f_v$ have same formulation. Loading can follow the Scatter or Gather pattern. Both patterns interpret tokens into a grid shape and find corresponding needed tokens using $f$;

Compute attention based on the shared memory;

Store the attention result back to $O$ in global memory using $f_o$;

---

## D  ARBITRARY SPARSE PATTERN COROLLARY

Given an arbitrary sparse pattern, we typically cannot formulate it into well-structured rectangular regions on the sparse attention map. Even using the INTRA kernel implementation, we still need to mask the extra-calculated attention scores to create the desired pattern. Therefore, we could turn the arbitrary pattern implementation into an optimization problem of choosing the least number of Computational Query Sets while ensuring we have the least number of masking operations. We formulate our corollary as follows:

Given an arbitrary sparse pattern, assume we want to choose all the CQSs greedily. For choosing a single CQS $\mathcal{S}$, we define a function $\text{Col}(s_i)$ that maps the query token $s_i$ in the current CQS to the number of columns it needs to calculate inside the sparse attention map. For a CQS, the total columns we need to calculate is

$$|\mathbb{S}| \cdot |\bigcup_{s_i \in \mathbb{S}} \text{Col}(s_i)|$$

The total columns that do not need masking is

$$\sum_{s_i \in \mathbb{S}} |\text{Col}(s_i)|$$

We need to find a $\mathbb{S}$ such that

$$\max_{\mathbb{S}} \quad \frac{\sum_{s_i \in \mathbb{S}} |\text{Col}(s_i)|}{|\mathbb{S}| \cdot |\bigcup_{s_i \in \mathbb{S}} \text{Col}(s_i)|}$$

The function $k(\mathbb{S}) = |\bigcup_{s_i \in \mathbb{S}} \text{Col}(s_i)|$ is a submodular function. If we have a CQS $\mathbb{M}$, for any subset $\mathbb{N} \subseteq \mathbb{M}$ and element $v \in \mathbb{M} \setminus \mathbb{N}$, $k(\mathbb{N} \cup \{v\}) \geq k(\mathbb{M} \cup \{v\})$. The function $\frac{\sum_{s_i \in \mathbb{S}} |\text{Col}(s_i)|}{|\mathbb{S}|}$ is also a submodular function. To maximize the ratio between two submodular functions, we follow the solution framework proposed in Bai et al. (2016). The selection of all CQSs can then be approached either via a greedy algorithm guided by this ratio-based optimization objective, or by formulating the task as a token partitioning problem.

# E   THEORETICAL ANALYSIS OF PATTERN INTERLEAVING AND INFORMATION PROPAGATION

Sparse attention restricts each token to attend only to a subset of other tokens. Therefore, for a sequence to achieve full information flow, the token must interact with others across several layers. In this section, we use directed graph to model the tokens interaction. We provide a formal definition of connectivity under a sparse attention pattern and analyze how Scatter and Gather interleaving enables global information propagation with minimal depth.

## E.1   ATTENTION GRAPH FORMULATION

Consider a sequence of tokens $\{x_1, x_2, \ldots, x_N\}$. A sparse attention pattern at layer $l$ can be represented as a directed graph:

$$G^{(l)} = (V, E^{(l)}), \quad V = \{1, \ldots, N\}, \quad (i,j) \in E^{(l)} \iff x_i \text{ attends to } x_j.$$

Full information propagation across layers is achieved if the union graph becomes strongly connected:

$$\bigcup_{l=1}^{L} G^{(l)} \quad \text{is strongly connected.}$$

We define the minimal connectivity depth as:

**Definition** (Connectivity Depth). For a given set of sparse patterns, define

$$d(i \to j) = \min L \quad \text{such that } j \text{ is reachable from } i \text{ using } \bigcup_{l=1}^{L} E^{(l)}.$$

The pattern connectivity depth for the sparse pattern set is

$$D = \max_{i,j} d(i \to j).$$

## E.2   CONNECTIVITY OF SCATTER–GATHER INTERLEAVING

Our approach uses Scatter and Gather patterns and therefore there are two types of layers:

**Scatter Layer (S):**   Tokens are partitioned into disjoint groups, and perform attention within each group:

$$G^{(S)} : x_i \text{ attends only to tokens in group } G_{k(i)}, \quad k(i) = \text{tokens in } x_i\text{'s group}$$

This induces global connectivity inside each group.

**Gather Layer (G):**   Each token attends to a local neighborhood (larger than the Scatter group size) in token space:

$$G^{(G)} : x_i \text{ attends to tokens in } \mathcal{N}(i), \quad \mathcal{N}(i) = \text{indices adjacent to } i.$$

This ensures cross-group connectivity, namely:

$$\forall \text{ group } k, \quad G_k \cap \mathcal{N}(i) \neq \varnothing.$$

**Resulting Connectivity.**   Let $G^{(S)} \cup G^{(G)}$ denote the union connectivity graph of two layers. Since $G^{(S)}$ ensures full connectivity *within* a group, and $G^{(G)}$ connects each token to (at least) one token from *every other group*, the union graph becomes strongly connected in two layers:

$$D_{\text{Scatter+Gather}} = 2.$$

### E.3  TOWARD PATTERN OPTIMALITY

Based on this formulation, we propose two criteria for evaluating sparse attention patterns:

1. **Connectivity depth** $D$: lower $D$ implies faster global information propagation;
2. **Index computation complexity**: patterns with simpler index computation lead to lower kernel overhead and higher throughput.

Scatter–Gather achieves:

$$D = 2 \quad \text{with} \quad O(1) \text{ index computation per token,}$$

thus balancing theoretical connectivity and practical deployment efficiency.

## F  SPEED–QUALITY TRADE-OFF ANALYSIS

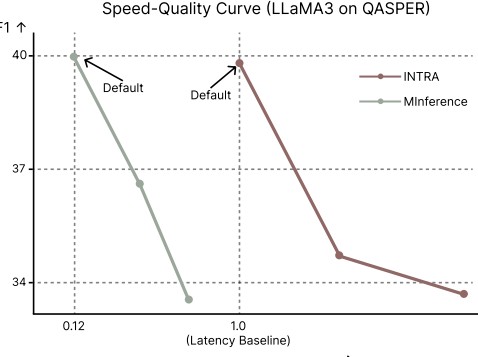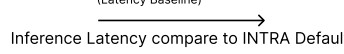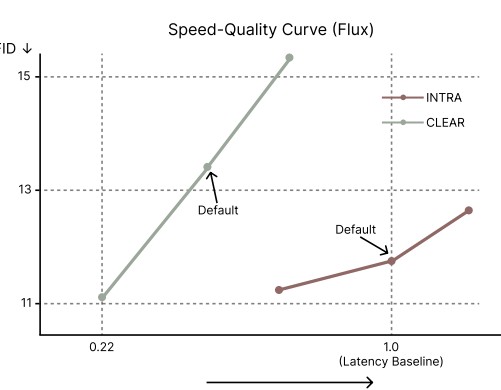

Figure 8: The speed quality trade-off curve for LLaMA3 and Flux.

We analyze how varying sparsity levels influence the efficiency–accuracy trade-off across two representative model families: LLaMA-3 (QASPER) and Flux (FID). For each task, we evaluate three configurations of the group number for INTRA, which serves as a sparsity control knob. A larger group number induces higher sparsity, resulting in lower computation and faster inference.

For LLaMA-3, we experiment with group sizes of 8 (default), 12, and 16, while for Flux, we use 4, 8 (default), and 16. We designate INTRA's default configuration (group size = 8) as the latency baseline for both tasks and normalize inference speed accordingly. This allows all methods to be compared fairly on the same operating scale.

The resulting speed–quality operating curves are shown in Figure 8. Across both models, INTRA demonstrates monotonic latency reduction as sparsity increases, confirming that INTRA effectively trades computation for efficiency without destabilizing accuracy. Both MInference (LLaMA-3) and CLEAR (Flux) exhibit similar latency scaling trends; however, under matched speed levels, their performance consistently falls below that of INTRA. Notably, INTRA matches the best performance in both tasks while operating at comparable or lower latency.

These results suggest that INTRA does not simply provide an alternative sparsity pattern—it shifts the Pareto frontier toward faster inference at equivalent quality levels. This demonstrates its potential as a deployment-efficient sparse attention mechanism with strong generalization across modalities.

## G  INTERELEAVING FREQUENCY

INTRA interleaves two distinct sparse attention patterns across model layers. To investigate how the interleaving frequency affects performance, we conduct an ablation study summarized in Table 6. The number of layers in the table indicates how many consecutive layers share the same sparse pattern before switching to the other. For example, when using a frequency of 2 layers, layers 0–1 apply the

Table 6: Ablation study on interleaving frequency of sparse patterns.

| Interleaving Frequency | FID ($\downarrow$) | LPIPS ($\downarrow$) | CLIPI ($\uparrow$) |
|---|---|---|---|
| 1 layer | 12.6 | 51.7 | 0.895 |
| 2 layers | **11.7** | **49.4** | **0.904** |
| 3 layers | 13.1 | 51.0 | 0.892 |
| 4 layers | 12.8 | 53.2 | 0.890 |

Table 7: Original LLaMA 3.1 8B model's performance on SCROLL benchmark without finetuning.

| | Gov Report | Qasper | Qmsum | Quality | SummScreenFd |
|---|---|---|---|---|---|
| Original model(No fine-tune) | 24.13 | 34.91 | 21.17 | 0.0 | 16.77 |
| Original model | **31.05** | **40.09** | **25.37** | 26.0 | **20.51** |
| INTRA | 27.77 | 39.64 | 24.45 | **29.0** | 19.71 |

Scatter pattern, layers 2–3 apply the Gather pattern, and so on. We find that interleaving the Scatter and Gather patterns every 2 layers achieves the best performance across all metrics.

## H    ORIGINAL MODEL WITHOUT FINETUNING RESULT

In Table 7 we show the comparison between the original LLaMA model without fine-tuning, the original LLaMA model with fine-tuning, and INTRA with fine-tuning. The original model's performance shows a significant improvement after finetuning. INTRA can approach the fine-tuned original model's performance (except Qasper), demonstrating INTRA's training effectiveness.

## I    QUALITATIVE RESULT

Please refer to Figure 9 and Figure 10 for more qualitative results comparison between the Original `FLUX.1-dev` model, INTRA, and CLEAR. INTRA preserves image quality and often demonstrates greater robustness compared to CLEAR

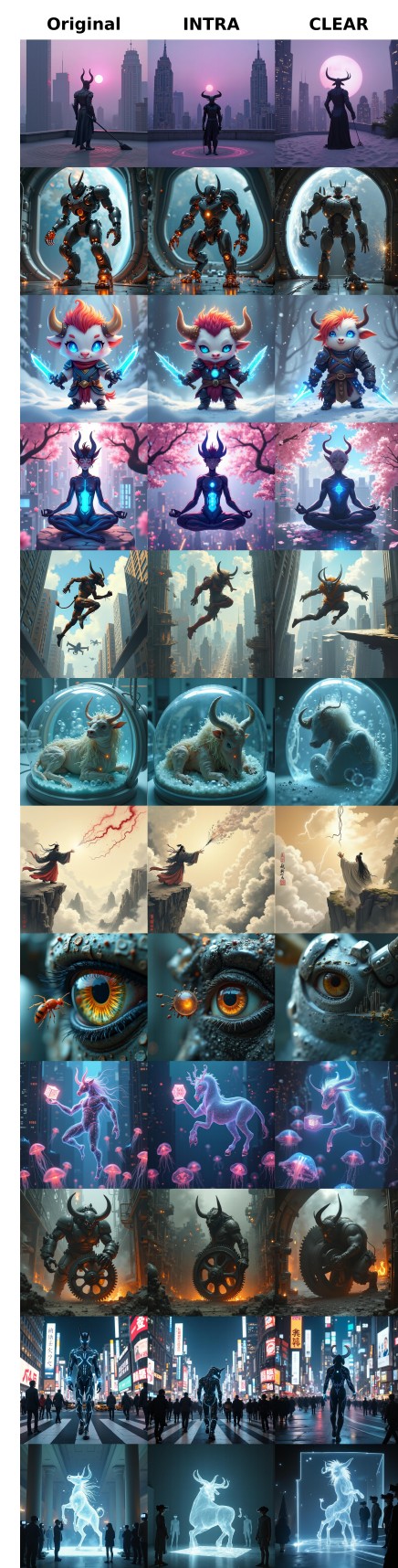
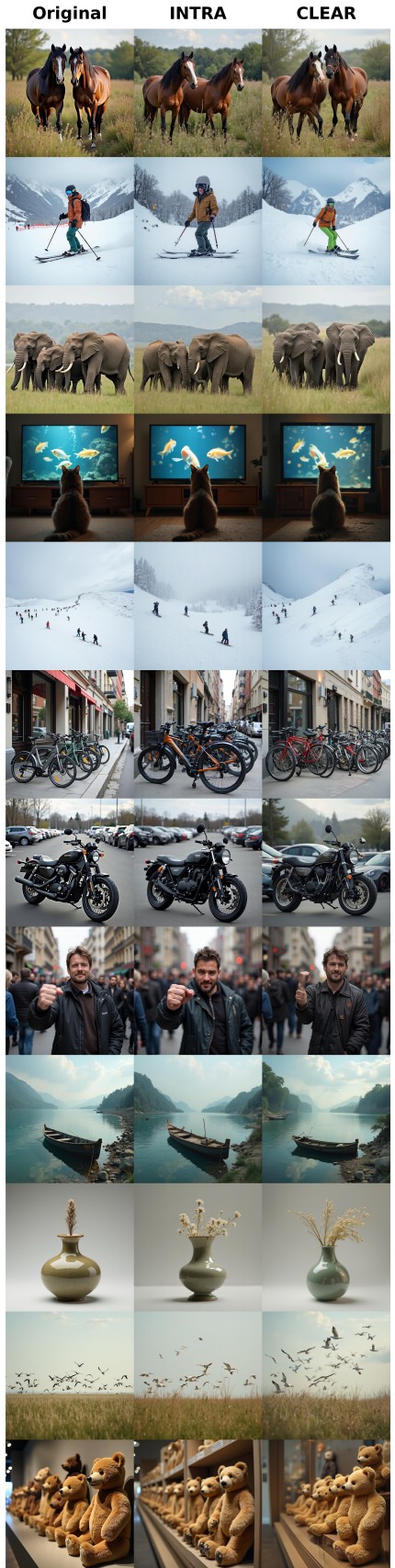

Figure 9: Visual examples comparison between original `FLUX.1-dev` model, INTRA, and CLEAR

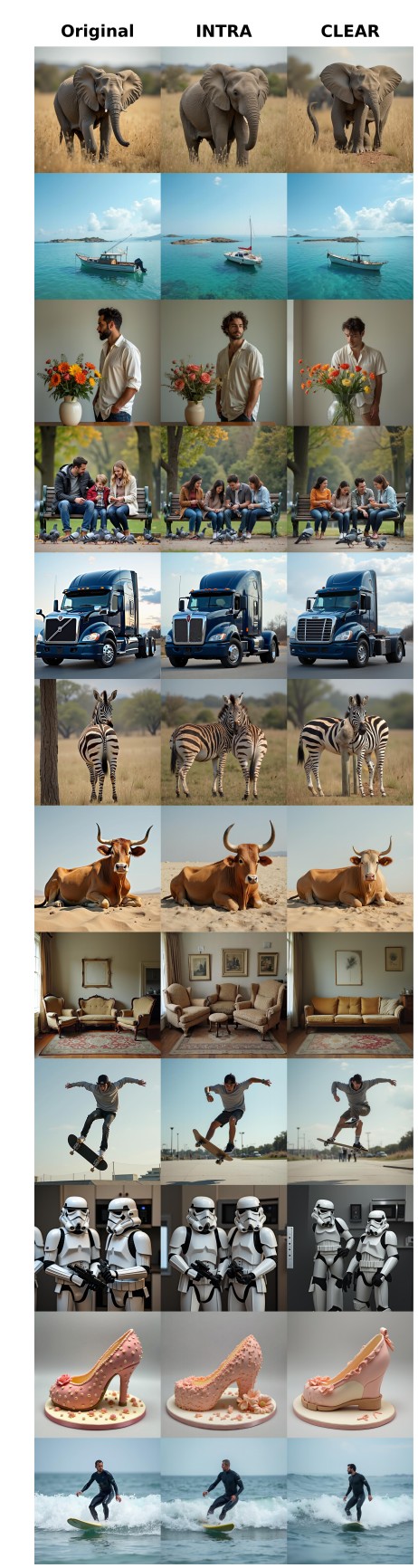

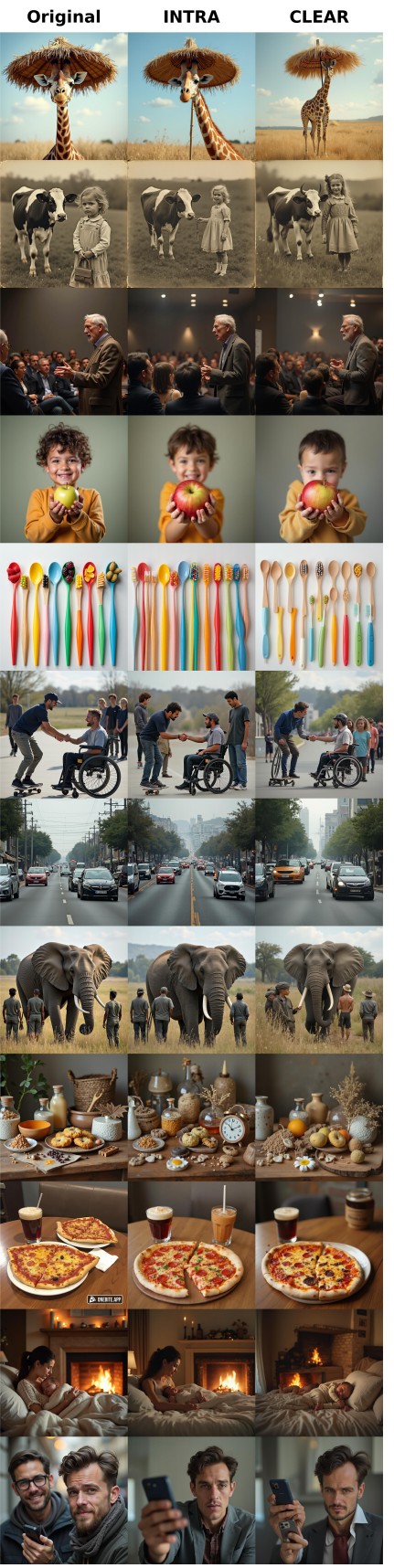

Figure 10: Visual examples comparison between original `FLUX.1-dev` model, INTRA, and CLEAR

