# OpenReview forum: "INTRA: Interleaved Non-contiguous Token spaRse Attention"
_ICLR.cc/2026/Conference — Submitted to ICLR 2026_

### Official Review · Reviewer_iMrY · 2025-10-31

**Soundness:** 3
**Presentation:** 3
**Contribution:** 2
**Rating:** 4
**Confidence:** 3

**Summary:**

The paper proposes to speed up attention with sparse grouping patterns. Query and keys are grouped into disjoint groups (attention complexity is quadratic in each group and linear in the number of groups). The method is simple to implement and allows to set a speed/accuracy trade-off by selecting the number of groups. Experimental comparisons with alternative sparse attention methods over text and image models are reported.

**Strengths:**

1. The proposed sparse attention method is simple: order (into groups) - attend - reorder and seems effective, even if the evaluation is partial (see weakness 1, 2).
2. The paper is well motivated, references to related literature are appropriate.
3. Experiments over image generation and text with long context are presented.

**Weaknesses:**

1. The reader cannot understand the speed/accuracy tradeoff in the number of groups. All sparse attention models can operate between regular attention (dense, slower, more accurate, i.e. one single group for INTRA) and no attention (maximum sparsity, fastest, less accurate, i.e. <input length> groups for INTRA), the results should be reported as an operating curve speed/accuracy between the two extremes. Sparse attention baselines also have a similar knob (e.g. local window size in CLEAR, or sparsity stripe for XATTN) and their results should be compared on the same plot.
2. The reader does not know if sparse attention is better than picking a smaller model. The speed/accuracy curve should also show regular attention models with varying numbers of parameters (smaller = faster but less accurate).
3. The presentation of the design of the attention groups is unclear. Could you show how “good” vs “bad” grouping pattern impact accuracy? Or speed? Or are all grouping patterns performing similarly?

**Questions:**

1. Timings are difficult to understand.
   1. Could you report generation timings for the non-attention part of the network (replacing attention with a dummy operation)?
   2. Are the speed-ups dependent on batch size?
   3. For text generation, could you report separately timings for prompt inference and autoregressive inference? Similarly, what is the network speed without attention?
2. Could you report FIDs results for the flashattn2 in Table 1, the original model in Table 3? Could you compare the FIDs with dense models of small size?
3. Table 5. Why did you drop two test sets from SCROLL? (ContractNLI and Narrative QA).
4. Could you also compare theoretical FLOPs per inference for the different models?
5. Have you considered methods that learn the grouping of query and keys like routing transformers https://aclanthology.org/2021.tacl-1.4/?
6. You say that methods with dynamic grouping e.g. https://arxiv.org/abs/2203.03937 are more complicated than static grouping but it would be good to report how well they perform on the speed / accuracy curve. Are static methods comparable?

---

> ### Author Response · Authors · 2025-11-26
>
> 1. **"The reader cannot understand the speed/accuracy tradeoff in the number of groups.”**
>    We appreciate this insightful observation. We added the Appendix F section on speed–quality trade-off analysis. The results confirm that INTRA scales smoothly with sparsity, producing a monotonic latency reduction without destabilizing model quality, unlike some baselines whose accuracy drops more sharply. On both Flux and LLaMA-3, INTRA reaches the same accuracy as the best baseline configurations while operating at lower cost, indicating that INTRA shifts the operating frontier toward the efficient regime. These trends are discussed in the Appendix, with full visualizations provided in Figure X.
> 2. **“The reader does not know if sparse attention is better than picking a smaller model.”**
>    We appreciate this important question. To address it, we compared INTRA with Stable Diffusion 1.5, a widely used smaller dense model on A100. The results demonstrate that sparsity alone is not equivalent to downscaling model capacity—INTRA significantly outperforms the smaller dense model. This suggests that sparse attention can indeed provide higher efficiency *without sacrificing model expressiveness*.
>
>
>     |  | FID ↓ |
>     | --- | --- |
>     | Stable Diffusion 1.5 | 30.02 |
>     | INTRA | 11.75 |
> 3. **“The presentation of the design of the attention groups is unclear…”**
>    Thank you for pointing out the need for additional clarity. We have added a new theoretical section (Appendix Section E) that provides a formal analysis for the interleaving sparse pattern designs. Here we provide a concise illustration:
> Two consecutive layers of INTRA ensure full token connectivity by designing the second-layer groups to contain tokens from all groups in the previous layer. While two layers are used in our implementation, it is not a strict requirement—other designs may use more layers. In general, patterns that achieve full connectivity with fewer layers and simpler index computation tend to provide better efficiency.
> 4. **“Timings are difficult to understand…”**
>    1. We report the full generation latency and the latency when attention is replaced with a dummy operation for the Flux model under 4K and 16K sequence length:
>
>
>         |  | 4K | 16K |
>         | --- | --- | --- |
>         | Flux Org | 13.4s | 66.8s |
>         | Flux remove attn | 11.6s | 41.2s |
>    2. **“Are the speed-ups dependent on batch size?”**
>    The speed-ups are not dependent on batch size, as INTRA operates orthogonally to batch parallelism in GPU kernels. Our method accelerates the full generation pipeline for diffusion models, and the only prefill stage for LLM, these two are all computationally bound.
>    3. **“For text generation, could you report separately timings for prompt inference and autoregressive inference…”**
>    In the experiment, we use INTRA only to accelerate the prefill stage, which may be referred to as “prompt inference” as mentioned. All the methods we are comparing for LLM are also prefill-only sparse attention methods. The decoding stage (autoregressive inference) is memory-bound, and sparse attention does not give significant speedups as computation is not the bottleneck. The corresponding prefill latency on A100 without attention is listed below.
>
>
>         |  | 4K | 8K | 16K | 32K |
>         | --- | --- | --- | --- | --- |
>         | Llama3 Org | 0.31s | 0.64s | 1.41s | 3.47s |
>         | Llama3 remove attn | 0.27s | 0.53s | 1.05s | 2.07s |
> 5. **“Could you report FIDs results for the flashattn2 in Table.."**
>    Thanks for the question. To clarify, the FID is computed against the original FLUX outputs, which already use FlashAttention. Therefore, reporting FID for FlashAttention-2 would compare the model against itself and is not meaningful. This task basically shows that by changing to INTRA, the model’s generation distribution does not change significantly. Instead, we provide a comparison against smaller dense models (see response 2).

---

> > ### Author Response · Authors · 2025-11-26
> >
> > 6. **“Table 5. Why did you drop two test sets from SCROLL? (ContractNLI and Narrative QA).”**
> >    Thanks for your question. We add the results on ContractNLI to Table 4. The Narrative QA has sequence lengths significantly beyond 16K tokens, requiring resources beyond our available hardware. We initially did not include ContractNLI because it lacks a consistent prompt setting, and related methods (MInference, xAttn) also exclude it.
> >
> > 7.  **“Could you also compare theoretical FLOPs per inference for the different models?”**
> >    Thanks for your question. We updated the Table 2 in the paper for MInference and xAttn for your reference.
> >
> > 8. **“Have you considered methods that learn the grouping of query and keys…”**
> >    That’s a very good point, and we agree that learning is possible. One point we want to note is that If the learned patterns satisfy the ISPD constraints, they can yield efficient execution; otherwise, the grouping creates a complex constrained optimization problem and loses kernel efficiency. We consider this a promising future direction.
> > 9. **“You say that methods with dynamic grouping…”**
> >    Thank you for pointing this out. These dynamic-grouping methods were developed before FlashAttention and target vision-specific tasks. They are not hardware-aware, and their kernels do not support efficient GPU execution under FlashAttention-style computation. Additionally, they do not release implementation details, preventing fair comparison. INTRA instead focuses on simulating full attention under modern GPU constraints, which differentiates it fundamentally from earlier vision-oriented grouping methods.

---

> ### Author Response · Authors · 2025-11-28
>
> Dear Reviewer,
>
> We appreciate your valuable feedback and have carefully addressed all comments in detail. We have submitted our responses and are awaiting any follow-up feedback, as the rebuttal deadline approaches. Please let us know if further clarifications or supporting materials are required, we would be glad to provide them.

---

### Official Review · Reviewer_A4gL · 2025-11-01

**Soundness:** 3
**Presentation:** 4
**Contribution:** 2
**Rating:** 4
**Confidence:** 3

**Summary:**

This paper introduces INTRA, a sparse attention kernel. The key innovation is decoupling memory loading from computation, enabling efficient non-contiguous token access while preserving GPU blockwise efficiency. INTRA alternates between two complementary patterns across layers: "Scatter" (which divides tokens into groups attending within-group) and "Gather" (which enables cross-group information exchange). The authors propose the ISPD (Intra Sparse Pattern Design) Principle as a general framework for hardware-efficient sparse pattern design. Experimental results show significant speedups on FLUX.1-dev image generation (66s to 43s for 2K images) and competitive performance on LLaMA-3.1-8B with sequence lengths up to 16K on the SCROLL benchmark.

**Strengths:**

1. Novel kernel design approach: The decoupling of memory loading from computation is elegantly engineered.
2. Practical efficiency gains: Demonstrates decent speedups (up to 7.67× on 16K sequences) with minimal quality degradation.
3. Practical fine-tuning approach: Shows that LoRA adaptation is sufficient to adapt pretrained models to INTRA, making the method accessible without prohibitive retraining costs (43 GPU hours for FLUX, 320 for LLaMA).

**Weaknesses:**

1. Limited theoretical justification for pattern interleaving

The paper relies heavily on intuition from image subsampling (Fig.4) but lacks rigorous theoretical analysis of why Scatter/Gather interleaving preserves global information flow. While the ablation (Table 5a) shows empirical necessity, there's no formal characterization of:
a. How many layers of interleaving are required for full connectivity
b. What properties of the patterns guarantee information propagation
c. Whether certain pattern combinations are provably better than others

2. Inconsistent and concerning performance on Qasper

Table 4 shows INTRA achieving only 36.68 on Qasper vs. 39.94/39.99 for baselines which is a decent gap. The explanation in Appendix G attributes this to "data hunger" and dataset size (2,567 examples), but:
a. Other SCROLL tasks have similar or smaller training sets
b. No ablation investigates whether this is fundamental to the sparse pattern design
c. This raises concerns about generalizability to complex reasoning tasks.

3. Missing critical baselines and comparisons

No comparison with block-sparse methods that also maintain GPU efficiency
The comparison with CLEAR is limited (only on image generation), and CLEAR represents just one prior approach

4. Insufficient analysis of overhead sources

Section 6.3 mentions overhead from modulo operations and uncoalesced memory access but provides no quantitative breakdown:
a. What fraction of runtime is spent on index computation vs. memory access vs. computation?
b. How does this overhead scale with sequence length?
c. Table 2 shows lower speedups for Scatter vs. Gather 0, but no detailed profiling explains why

5. Limited scope of experimental validation

a. Maximum sequence length tested is 32K (Table 2), which is modest by current long-context standards
b. Authors acknowledge inability to test on video due to resources, but this is a key claimed application...
c. No experiments on domains like code generation, retrieval-augmented generation, or other long-context tasks beyond SCROLL
d. The 16K context length claim as "standard" is outdated (many recent models support 100K+)

6. Incomplete formalization of the ISPD Principle
The ISPD Principle (Section 4) is presented as a "general guideline" but:
a. Condition 3 (Block Alignment) uses "approximately divisible" without defining the approximation tolerance
b. The claim that "sparse patterns that have only intra-CQS attention are efficient" is not proven or formally connected to GPU performance
c. The optimization formulation in Appendix D is incomplete (no algorithm provided, just problem statement)

**Questions:**

Please address some implicit questions in the weakness.

Quality-efficiency trade-off

Tables 1 and 2 show GFLOPS reductions, but can you provide Pareto curves showing the quality-efficiency trade-off as you vary the number of groups or interleaving frequency? Where does INTRA sit relative to baselines?

Statistical significance

Are the differences in Table 3 and 4 statistically significant? Can you provide confidence intervals or statistical tests, especially for close comparisons (e.g., INTRA vs. CLEAR on 1K generation)?

---

> ### Author Response · Authors · 2025-11-26
>
> 1. **“Limited theoretical justification for pattern interleaving.”**
>    Thank you for raising this important point. We have added a new theoretical section (Appendix Section E) where we provide a formal analysis of connectivity under sparse attention patterns. Below, we provide concise answers to your specific concerns:
>
>    a. **“How many layers of interleaving are required for full connectivity…”**
>       In our current design, two consecutive layers of interleaving are sufficient to achieve full token connectivity. However, this is *not a strict requirement* — different pattern designs may use more layers depending on the grouping strategy and target sparsity level.
>
>    b. **“What properties of the patterns guarantee information propagation”**
>       The key property in our design is cross-group coverage: each token group in the second layer contains tokens from all groups in the previous layer. This ensures transitive mixing and enables global information flow.
>
>    c. **“Whether certain pattern combinations are provably better than others”**
>       Empirically, we observe that patterns requiring fewer layers for full connectivity and simpler index computation yield better efficiency. We view this as a first step toward formalizing pattern optimality, and we agree that this is an exciting direction for future theoretical extension.
>
>
> 2. **“Inconsistent and concerning performance on Qasper”**.
>    Thank you for highlighting this. We conducted additional experiments with a higher LoRA rank (32 vs. 8) and observed an improved score of 39.64 on Qasper, matching other baselines. This suggests that INTRA is not inherently limited to complex reasoning tasks, but rather benefits from increased adaptation capacity. These results strengthen INTRA’s generalizability across task types.
> 3. **“Missing critical baselines and comparisons”**.
>    We appreciate this insight. In the image generation domain, very few sparse approaches are applicable under modern GPU-aware implementations. Methods such as Swin Transformer and Strided Attention were proposed before FlashAttention and rely on less GPU-friendly kernels, resulting in significantly higher inference latency. As requested, we have included FID and GFLOPS comparisons with these methods for completeness.
> Regarding block-sparse attention, to the best of our knowledge, there are currently no publicly available implementations tailored to diffusion-based image generation with efficient GPU execution. We view this as an open research gap, and INTRA fills this need by providing a more efficient and compatible solution with recent architectures.
>
>
>     |  | FID | GFLOPS |
>     | --- | --- | --- |
>     | Strided Attention | 24.88 | 67.7 |
>     | Swin Transformer | 18.90 | 67.7 |
>     | CLEAR | 11.70 | 63.5 |
>     | INTRA | 11.74 | 50.75 |

---

> > ### Author Response · Authors · 2025-11-26
> >
> > 4. **“Insufficient analysis of overhead sources”**.
> >
> >     1.  **“What fraction of runtime is spent on index computation vs. memory access vs. computation?”**. We appreciate your request for a deeper analysis of the computational characteristics of our Scatter/Gather kernels. To address this, we conducted profiling using NVIDIA Nsight Compute and Nsight Systems. However, both tools inherently aggregate operations at the kernel level and do not provide precise isolation of index computation, memory access, and computation time as independent components. Despite this limitation, we provide profiling results that shed light on the behavior of Scatter/Gather compared to dense attention.
> >
> >         **Profiling Analysis**
> >
> >         The key observation is that our Scatter/Gather kernels exhibit notably higher memory throughput but lower compute throughput relative to Flash Attention:
> >
> >         |  | Memory Throughput | Compute Throughput | Actual Inference Time |
> >         | --- | --- | --- | --- |
> >         | Flash Attention | 49.95 | 70.69 | 15.8 ms |
> >         | INTRA Scatter | 86.31 | 56.64 | 3.04 ms |
> >         | INTRA Gather | 79.48 | 51.19 | 1.07 ms |
> >
> >         This suggests that Scatter/Gather shifts the bottleneck from computation to memory access. This is expected: while dense attention spends substantial time on compute-intensive QKᵀ operations, our sparsity patterns reduce these computations, causing the kernels to become memory-bound. The lower compute throughput thus does not indicate inefficiency; rather, it reflects the reduced arithmetic workload, consistent with the intended sparsity. Conversely, the higher memory throughput indicates that the kernels fully exploit GPU memory bandwidth, supporting efficient data movement under sparse patterns.
> >
> >
> >         **Simulation of Matching Sparsity**
> >
> >         We also simulate an “equally sparse” case by crafting inputs for dense Flash Attention that produce the same sparsity level as our Scatter/Gather patterns:
> >
> >         | Sequence Length | INTRA Scatter (ms) | Equally Sparse (ms) |
> >         | --- | --- | --- |
> >         | 4K | 0.17 | 0.19 |
> >         | 8K | 0.53 | 0.52 |
> >         | 16K | 1.79 | 1.65 |
> >         | 32K | 6.58 | 5.73 |
> >
> >         | Sequence Length | INTRA Gather (ms) | Equally Sparse (ms) |
> >         | --- | --- | --- |
> >         | 4K | 0.29 | 0.22 |
> >         | 16K | 1.07 | 0.79 |
> >
> >         This comparison provides a rough reference for overhead. However, we stress that this simulation cannot reflect realistic inference conditions. Crafting input shapes for sparsity alters the kernel launch configuration (e.g., block size, occupancy), which may accidentally increase parallelism or lead to suboptimal scheduling. Thus, this experiment only serves as an approximate sanity check rather than a conclusive overhead analysis.
> >
> >     2. **“How does this overhead scale with sequence length?”**
> >    Index computation scales *linearly* with sequence length.
> >     3. **“Table 2 shows lower speedups for Scatter vs. Gather 0, but no detailed profiling explains why.”**
> >    Thanks for your question. Based on our Nsight Compute profiling result. Gather 0 has a more contiguous memory loading pattern, which leads to much less uncoalesced memory loading in global memory. This is the reason for the slightly lower speedup in the scatter pattern.
> > 5. **“Limited scope of experimental validation”**.
> >    Thank you for pointing this out. Our current limit of 32K sequence length is due to hardware constraints during fine-tuning. We agree that very long contexts (100K+) are increasingly important. INTRA is designed to be compatible with longer contexts, and we expect larger gains as sequence length increases — but verification requires hardware that exceeds our current capacity. Especially for video generation, it’s straightforward to implement 3D Scatter/Gather patterns. However, due to resource constraints, we could not fine-tune video models to report full generation metrics. We consider this an important next step.

---

> ### Author Response · Authors · 2025-11-26
>
> 6. **“Incomplete formalization of the ISPD Principle…”**
>    a. **“Condition 3 (Block Alignment) uses "approximately divisible" without..”**
>    We use approximately divisible because we don’t want to waste computation. If the sequence length cannot be divisible by the block size of 64, the last block will waste some computation. However, in practice, this overhead is negligible in the long-context generation. First, the original dense Flash Attention also wastes the last block if the remaining sequence is less than 64. Second, the waste is at most 64 tokens, which is very insignificant compared to the full calculation of over 10K sequence. We stress this point just to state the most optimal design.
>    b. **“The claim that "sparse patterns that have only…”**
>    Thanks for your question. Allowing inter-CQS computation requires additional synchronization across thread groups to collect outputs before final storage. This introduces nontrivial overhead and needs plenty of modifications on FlashAttention kernels. Restricting to intra-CQS computation preserves GPU efficiency and kernel parallelism.
>    c. **“The optimization formulation in Appendix D is incomplete…“**
>    We follow the solution framework in Bai et al. (cited in line 808). A full algorithmic derivation is beyond our current scope but represents a promising extension.
> 7. **“Quality-efficiency trade-off…”**
>    Thank you for the suggestion. We have now added the quality–efficiency trade-off analysis in Appendix F. Across both tasks, the operating curves show that INTRA lies on or beyond the Pareto frontier: it achieves equal or higher quality than MInference and CLEAR at comparable speed, and it reaches their best reported quality while operating at a lower latency. This demonstrates that INTRA does not simply reduce GFLOPS but provides a better trade-off between inference efficiency and model accuracy. The results are summarized in Figure 8 in Appendix F.
> 8.  **“Statistical significance…”**.
>    Thank you for raising this point.
>     - **Latency**
>
>         All the inference time result in our paper is averaged over 2,000 runs. We provide the variance measurements for generation time to assess statistical significance across multiple resolutions and prefill lengths in the table below. The variances demonstrate that the latency improvements of INTRA are stable and statistically significant, showing that our efficiency gains are not due to noise or randomness.
>
>     - **Benchmark Scores**
>
>         For benchmark results in both image generation and long context text generation, the performance differences between INTRA and baselines are small and not statistically significant.

---

> ### Author Response · Authors · 2025-11-28
>
> Dear Reviewer,
>
> We appreciate your valuable feedback and have carefully addressed all comments in detail. We have submitted our responses and are awaiting any follow-up feedback, as the rebuttal deadline approaches. Please let us know if further clarifications or supporting materials are required, we would be glad to provide them.

---

### Official Review · Reviewer_HXkx · 2025-11-02

**Soundness:** 3
**Presentation:** 3
**Contribution:** 3
**Rating:** 6
**Confidence:** 3

**Summary:**

The paper proposes INTRA (Interleaved Non-contiguous Token Sparse Attention), a static, token-level sparse attention kernel that decouples memory loading from attention computation. By enforcing “intra-CQS” (computational query set) attention, the kernel keeps GPU-friendly blockwise access while still supporting non-contiguous token sparsity—avoiding the mixed-block masking overhead common in CLEAR-style block sparsity. To restore global information flow that a single sparse pattern cannot provide, the model interleaves complementary Scatter and Gather patterns across layers. The authors further generalize this into the ISPD Principle, a recipe for designing hardware-efficient sparse patterns. On FLUX.1-dev, INTRA cuts 2K image generation latency 66s → 43s without quality loss (with LoRA self-distillation); on LLaMA-3.1-8B, INTRA attains comparable SCROLL long-context quality up to 16K while giving 5–7× single-attention speedups over FlashAttention 2.

**Strengths:**

1. **Hardware-aligned idea.** The key insight “only tokens in the same CQS attend to the same keys” is a good GPU sight.

2. **Interleaved patterns actually matter.** Ablation shows Scatter-only or Gather-only collapses quality, while alternating recovers FID/LPIPS/CLIP to dense-like levels.

3. **Cross-modality evidence.** Vision (FLUX.1-dev, 1K/2K) + LLM (SCROLL 16K/32K), showing that method is not overfitted to one workload.

**Weaknesses:**

1. **Limited LLM task coverage.** Performance on Qasper is clearly worse; authors only hypothesize it’s data-hungry. A small controlled study (more LoRA steps / full FT on one QA set) would make the claim stronger.
2. **Overhead not fully quantified.** Section 6.3 admits modulo + uncoalesced loads hurt speed; paper should give a table: “ideal vs actual” for Scatter/Gather separately.
3. **Static-pattern requirement.** Method still needs finetuning to adapt to the static pattern; cannot be dropped into arbitrary checkpoints like dynamic methods.
4. **Vision eval is narrow.** Only FLUX.1-dev (one heavy diffusion transformer). A 1B/2B open-image model or a video-lite model would help the “general” claim.

**Questions:**

1. For Qasper: if you double the LoRA rank or increase training steps, does the gap shrink?
2. Can INTRA be combined with head-level specialization (e.g., some heads always Gather, some always Scatter) to reduce interleaving frequency without losing connectivity?
3. You mention token-wise loading “generalizes” FlashAttention. What is the max sparsity irregularity you can handle before shared-memory packing becomes the bottleneck?
4. For video: since you already have 2D Scatter/Gather, can you simply stack a temporal Gather every N layers to get 3D coverage, or does ISPD break because of block-size mismatch?

---

> ### Author Response · Authors · 2025-11-26
>
> 1. **“Overhead not fully quantified…”**
>    Thank you for the suggestion. We have added a table reporting actual vs. ideal speedup for both Scatter and Gather on language models. Tokens are divided into 8 groups, so the theoretical speedup is 8×. As shown below, the actual speedup increases with sequence length, meaning the relative overhead becomes smaller for longer-context inference:
>
>
>     | Pattern | Sequence Len | Ideal Speedup | Actual Speedup |
>     | --- | --- | --- | --- |
>     | Scatter | 4K | 8 | 4.81 |
>     | Gather | 4K | 8 | 5.84 |
>     | Scatter | 8K | 8 | 5.43 |
>     | Gather | 8K | 8 | 5.83 |
>     | Scatter | 16K | 8 | 5.90 |
>     | Gather | 16K | 8 | 6.15 |
>     | Scatter | 32K | 8 | 6.22 |
>     | Gather | 32K | 8 | 6.37 |
>
> 2. **“For Qasper: if you double the LoRA rank or increase training steps, does the gap shrink?”**
>    We appreciate the suggestion. We conducted additional experiments by increasing the LoRA rank from **8 → 32**. With this adjustment, the Qasper accuracy improved to **39.64**, closing the gap and aligning closely with both MInference and xAttn. This confirms that INTRA’s effectiveness in finetuning.
> 3. **“Can INTRA be combined with head-level specialization…”**
>    Yes, INTRA can be combined with head-level specialization, where certain heads exclusively perform Scatter or Gather. This strategy is compatible with GPU computation. However, it requires grouping heads during fine-tuning, introducing additional training complexity. While we decided not to pursue this direction in our main method, we agree that it is a promising direction for future exploration.
> 4. **“You mention token-wise loading “generalizes” FlashAttention…”**
>    Thanks for the question. The shared-memory bottleneck arises mainly from index computation, not from sparsity itself. Highly sparse patterns may actually yield simpler index calculations, and therefore, do not necessarily introduce much overhead. If the “irregularity” refers to the complexity of index computation, we think it is hard to quantify such complexity. In the ideal case, more sparsity is more efficient as long as the index computation can be performed efficiently as well.
>
> 5. **“For video: since you already have 2D Scatter/Gather, can you simply stack a temporal…”**
>    Thanks for your question. Yes — stacking a temporal Gather layer every N layers successfully extends our 2D mechanism to 3D, and does not break ISPD. Since video data is well-structured frame-by-frame, the index calculations for 3D Scatter/Gather are straightforward. We have already designed such sparse loading for video. However, as stated in the paper, we currently lack the ability to fine-tune a video model, preventing us from evaluating the method on real video generation benchmarks.

---

> ### Author Response · Authors · 2025-11-28
>
> Dear Reviewer,
>
> We appreciate your valuable feedback and have carefully addressed all comments in detail. We have submitted our responses and are awaiting any follow-up feedback, as the rebuttal deadline approaches. Please let us know if further clarifications or supporting materials are required, we would be glad to provide them.

---

### Author Response · Authors · 2025-12-03
**Author Response Summary to Reviewer Feedback**

We thank the reviewers for their constructive feedback. Below is a concise summary of how our additional experiments and analyses directly address the reviewers’ concerns to facilitate efficient assessment under the updated review process.

**Reviewer HXkx**

- **Overhead quantification:** Added a new table comparing ideal vs. actual Scatter/Gather speedups, showing overhead is small and decreases with sequence length.
- **Qasper performance:** Increasing LoRA rank (8→32) improves Qasper accuracy to **39.64**, nearly matching baselines.
- **Pattern flexibility:** Clarified that INTRA supports head-level specialization (fixed Scatter/Gather heads), offering flexibility beyond static patterns.
- **Sparsity irregularity:** Explained that overhead arises from index computation rather than sparsity. Sparse patterns with simple indexing remain efficient.
- **Video generality:** Showed how stacking temporal Gather layers extends INTRA naturally to 3D, conceptually supporting video models.

**Reviewer A4gL**

- **Theoretical justification of interleaving:** Added a formal analysis in Appendix E, clarifying full-connectivity conditions, coverage, and efficiency, addressing sub-questions (a–c).
- **Qasper performance:** Higher LoRA rank (32) raises accuracy to **39.64**, showing the effectiveness of INTRA.
- **Baselines:** Added comparisons with Swin and Strided Attention; clarified why block-sparse methods are incompatible with modern diffusion pipelines.
- **Overhead analysis:** Provided Nsight profiling, clarified why Gather is more efficient, and documented scaling under profiling constraints.
- **Experimental scope:** Justified the 32K limit and explained how INTRA extends to longer contexts and video models.
- **ISPD principle formalization:** Clarified block-alignment tolerance, GPU rationale for intra-CQS operations, and referenced the optimization framework.
- **Quality–efficiency trade-off:** Added Pareto curves (Appendix F), showing INTRA matches or surpasses the Pareto frontier.
- **Statistical significance:** Reported latency variance (2,000 runs) and clarified that benchmark differences are not statistically significant.

**Reviewer iMrY**

- **Speed–accuracy trade-off:** Added full operating curves (Appendix F), showing smooth sparsity scaling and INTRA’s superior efficiency relative to baselines.
- **Comparison to smaller dense models:** Added comparison with Stable Diffusion 1.5, showing INTRA maintains expressiveness while achieving far better FID.
- **Attention grouping design:**  Added formal analysis (Appendix E) and clarified how two-layer patterns ensure full connectivity and differ in efficiency.
- **Timing breakdowns:** Added attention-removed timings, clarified batch-size independence, and reported prefill latencies for LLMs.
- **FID completeness:** Explained why reporting FlashAttention-2 FID is not meaningful.
- **SCROLL evaluation completeness**: Added ContractNLI and clarified omission of NarrativeQA due to extreme length.
- **FLOPs comparison:** Updated FLOPs for MInference and xAttn, enabling clearer theoretical comparison.
- **Dynamic and learned grouping:** Explained inefficiency of dynamic routing under FlashAttention constraints and noted learned grouping as future work.

Overall, our additions address the reviewers’ concerns, strengthen both the theory and the empirical evidence, and reinforce the reliability and generality of INTRA.

---

### Meta-Review · Area_Chair_ZzVU · 2026-01-05

**Summary:**

Strength: The paper is well motivated, and the proposed method appears to be practically relevant and achieves efficiency gain.

Weakness: Reviewers have raised major concerns about theoretical justification of the method, and the experimental section, in particular, baseline comparison and limited scope of experimental validation.

**Reviewer Concerns:**

Weakness: Reviewers have raised major concerns about theoretical justification of the method, and the experimental section, in particular, baseline comparison and limited scope of experimental validation.

**Reviewer Scores:**

Based on the comments and discussion, the reviewers would likely keep their scores.

---

### Decision · Program_Chairs · 2026-01-26

Reject